# Longitudinal Effects of Motivation and Physical Activity on Depressive Symptoms among College Students

**DOI:** 10.3390/ijerph18105121

**Published:** 2021-05-12

**Authors:** Jie Zhang, Xiangli Gu, Xiaoxia Zhang, Jihye Lee, Mei Chang, Tao Zhang

**Affiliations:** 1College of Physical Education and Sports Training, Shanghai University of Sport, Shanghai 200438, China; zhangjie@sus.edu.cn; 2Department of Kinesiology, The University of Texas at Arlington, Arlington, TX 76019, USA; 3Department of Kinesiology, East Carolina University, Greenville, NC 27858, USA; zhangxi20@ecu.edu; 4Department of Health and Kinesiology, Texas A&M University, College Station, TX 77845, USA; vkstm49@gmail.com; 5Department of Educational Psychology, University of North Texas, Denton, TX 76203, USA; mei.chang@unt.edu; 6Department of Kinesiology, Health Promotion, and Recreation, University of North Texas, Denton, TX 76203, USA; tao.zhang@unt.edu

**Keywords:** depressive symptoms, psychosocial mechanism, physical activity motivation, physical activity

## Abstract

High prevalence of depression and physical inactivity have been consistently reported among college students, especially in females. Guided by Lubans et al.’s conceptual framework, the primary purpose of this study was to examine the longitudinal relationships of PA motivation with leisure-time PA and depressive symptoms among college students over one academic year. Employing a longitudinal repeated measure design, 1004 college students in China were recruited in this study (28.3% males and 71.7% females; *M* age = 18.93 ± 0.64 years; 18–22 years old). Participants completed previously validated questionnaires assessing PA motivation (perceived competence beliefs and task values toward PA), leisure-time PA participation, and depressive symptoms in Fall 2016 (Time 1) and Fall 2017 (Time 2). Both male and female college students showed a significant increase of depressive symptoms from freshmen to sophomores (*p* < 0.05). The regression models indicated that perceived competence beliefs and task values toward PA were significant predictors of depressive symptoms at Time 2 (*p* < 0.05) after controlling for Time 1 measures in males and females, respectively. Physically active college students consistently demonstrated higher PA motivation, and they displayed fewer depressive symptoms compared to inactive peers over time (*p* < 0.05). The findings suggest sex-specified motivational intervention strategies and PA promotion programs/opportunities are needed to reduce depression symptoms among college students over time.

## 1. Introduction

Depression has become one of the leading causes of mental health illnesses worldwide and one of the most common mental disorders among college students [1,2]. Approximately 20% of college students in the United States were diagnosed with depression [3] and a similar prevalence rate (23.8%) was also reported among Chinese college students [4]. According to World Health Organization’s definition, depression is defined as “an illness characterized by persistent sadness and a loss of interest in activities, accompanied by an inability to carry out daily activity” and the depressive symptoms could last for at least two weeks [5]. It is well-documented that participation in regular physical activity (PA) plays an important role in the prevention of developing depressive symptoms among adults [6,7,8]. Growing research noted that, however, people with depressive symptoms often do not meet daily PA recommendations [9,10], and college students who show depressive symptoms are less likely to participate in regular PA compared to their counterparts without depressive symptoms [10,11].

The 2018 PA guideline for Americans recommends that adults aged 18–64 years old should participate in at least 150 min of moderate PA (3–6 metabolic equivalents—METs; e.g., fast walking, easy swimming), 75 min of vigorous PA (over 6 METs; running, basketball), or an equivalent combination of moderate-to-vigorous PA (MVPA; over 3 METs) every week [12]. In order to obtain additional health benefits, they also need to do muscle-strengthening activities at least 2 or more days per week [12]. In 2016, China released the National Fitness Program to elevate the policy prominence of sport for all, which recommends daily 30–60 min moderate PA [13]. However, only roughly one-thirds (34%) of the Chinese college students achieved the PA recommendation, and they often developed unhealthy lifestyles during the transition from adolescence to adulthood, such as being physically inactive/sedentary and engaging in less outdoor activities [14]. Research also reported that first year college students experience reduced family support, increased relational instability, and climbed financial and academic stress [15,16]. Loss of interest in physical activities, feelings of failure, and decreased energy are commonly reported symptoms among individuals who are depressed [5]. A recent review of 19 longitudinal cohort studies suggested that theoretical understanding of the psychosocial mechanism pertaining to depressive symptoms is urgent due to the inconsistent findings of relationships between PA and mental health outcomes [17].

Lubans and his colleagues proposed a conceptual model in understanding the relationship between PA and mental health outcomes with three potential mechanisms (i.e., neurobiological, psychosocial, and behavioral) [18]. According to the psychosocial mechanism, an individual’s self-perceptions (e.g., motivation toward PA and physical self-perception) may interact with PA behaviors to influence one’s mental health outcomes (e.g., depressive symptoms, quality of life). Bauman and colleagues reported that high levels of PA motivation, such as self-efficacy, were associated with high levels of PA among adults [19]. Additionally, a strong sense of autonomy and interest in PA were also found to be the significant correlates of PA and sedentary behavior among college students [20,21,22]. In a mix-methods study, for example, Otundo and MacGregor found that college students were motivated to participate in PA when they perceived autonomous support toward PA, enjoyment, and competence [22]. White et al. demonstrated preliminary longitudinal evidence that increased PA and self-efficacy may result in decreased depressive symptoms among individuals with elevated depressive symptoms [23]. However, the sample size of this longitudinal study was relatively small (*n* = 39), and the data were repeatedly measured in a short duration (8 weeks). To date, the longitudinal evidence on the relationship between PA and depressive symptoms has not been well-established, thus the potential psychosocial correlates in this association warrants further investigation.

Given the prevalence of depressive symptoms among college students, longitudinal research in this area that incorporate stronger research design elements (e.g., larger sample sizes, controlling confounding variables, longer measuring time frames) could provide invaluable insights for preventing and/or alleviating depressive symptoms in college students from a public health perspective. In addition, sex differences in PA and depressive symptoms were consistently observed, where males appeared more active and demonstrated less depressive symptoms compared to females counterparts [1,3,4,16]. Guided by Lubans and colleagues’ conceptual framework [18], the present study aimed to investigate (1) the longitudinal changes in PA motivation (perceived competence beliefs and task values toward PA), leisure-time PA participation, and depressive symptoms, and (2) the longitudinal associations between PA motivation, leisure-time PA participation, and depressive symptoms in a group of Chinese college students over a one-year period.

## 2. Materials and Methods

### 2.1. Participants and Settings

The present study employed a longitudinal repeated measure design over a 12-month period. An initial 1163 college students were recruited from two universities in China. Data collection was conducted in Fall 2016 (Time 1/Freshmen) and Fall 2017 (Time 2/Sophomores). Eventually, with a retention rate of 86.3%, 1004 students completed all assessments at both Time 1 and Time 2 points. Thus, a final sample of 1004 Chinese college students (71.7% females; mean age at Time 1 = 18.93 ± 0.64 years, 18–22 years old) were included in the data analysis. Distributions of participants’ college majors are Humanities (38.1%), Social Science (39.2%), Applied Science (14.4%), and Arts (8.3%). The demographic characteristics of the samples are shown in the Table 1.

### 2.2. Procedures

Approval for conducting the study was obtained from the University Institutional Review Board (IRB#16-588). Permission for data collection was granted by the administrators and the instructors of the PA classes at the two universities in China. In Fall 2016, college students from two universities were invited to participate in this study. Participants’ signed consent forms were obtained prior to the data collection. The data collection was administered by the trained research assistants in the physical education classes. Researchers introduced the objectives of the project and made sure students understand their participation is voluntary and they may quit the study at any time without negative consequences. After consenting process, participants completed the paper-pencil version of surveys in one sitting by self-reporting their PA motivation (PA competence beliefs and PA values), leisure-time PA, and depressive symptoms. The Time 2 measures were processed at the beginning of their sophomore year and followed the same protocol.

### 2.3. Measures

#### 2.3.1. Depressive Symptoms

The 10-item Chinese version of the Center for Epidemiologic Studies Depression Scale (CES-D) was used to assess participants’ depressive symptoms status [24,25]. The Chinese version of the CES-D has been utilized to investigate Chinese college students in a previous study, which reportedly has an acceptable internal consistency (α = 0.71) [25]. Participants were asked to respond to the frequency of each depressive symptom they experienced in the past week based on a 4-point Likert scale, ranging from 0 (Rarely or Not at all—<1 day) to 3 (Most or All of the time—5–7 days). The total score ranges from 0 to 30, with higher scores indicating higher likelihoods of being depressed.

#### 2.3.2. Physical Activity (PA) Motivation

The two PA motivation variables—perceived competence beliefs and task values toward PA—were measured by the Chinese-version of the Expectancy-Value Questionnaire (EVQ) [26], originally developed by Eccles and Wigfield [27]. The PA competence beliefs were measured by 5 items, using 5-point Likert descriptors (1 = Very bad, 5 = Very good). Sample items include: “How good are you at physical activities?” and “How well do you think you will participate in physical activities?” The mean value of the five items provided an overall indication of the magnitude of a student’s PA competence beliefs. The PA values were measured by 6 items, using 5-point Likert descriptors. Sample questions include (a) For me, being good at physical activities is… (1 = Not very important, 5 = Very important); (b) In general, I find learning new physical activities is… (1 = Very boring, 5 = Very fun); and (c) In general, how useful is what you learn in PA? (1 = Not useful at all, 5 = Very useful). The mean value of the six items was used to reflect students’ PA values. Internal consistency of the PA motivation scale was good for both Time 1 and Time 2 measures in the current study (PA competence beliefs: Time 1 α = 0.82, Time 2 α = 0.84; PA values: Time 1 α = 0.88, Time 2 α = 0.90).

#### 2.3.3. Leisure-Time Physical Activity (Leisure-Time PA)

The Leisure Time Physical Activity Scale was used to capture the intensity level of Leisure-time PA [28,29]. Participants recalled how many times they undertook the listed physical activities during the last 7 days, including strenuous (e.g., running, basketball, vigorous long-distance bicycling), moderate (e.g., fast walking, baseball, folk dancing), and mild (e.g., yoga, archery, golf) activities. The weighted Leisure-Time Score Index (LSI) was obtained using the LSI formula: (frequency of mild  ×  3)  +  (frequency of moderate  ×  5)  +  (frequency of strenuous  ×  9). The intended cut-point values for the classification scorings are based on the North American public health PA guidelines that are defined as follows: individuals reporting moderate-to-strenuous LSI  ≥  24 are classified as active, whereas individuals reporting moderate-to-strenuous LSI  ≤  23 are classified as insufficiently active (estimated energy expenditure  <  14 Kcal/kg/week) [29]. Good test–retest reliability of this scale has been found in a past study among college students, with coefficients ranging between 0.70 and 0.91 [30]. This measure has also been validated with objective measures of PA such as accelerometers and VO_2_max [31,32].

### 2.4. Data Analyses

IBM SPSS Statistics Version 25.0 (IBM Corp., Armonk, NY, USA, 2017) was used for data analyses. According to the power analysis, a minimum sample of 752 is required to detect a small effect size (ƒ = 0.10) of a repeated within group effect based on a statistical power of 0.99 with a probability level of 0.01. The sample size of 1004 in the current study was sufficient to carry out the data analysis. Before the formal data analysis, we screened the data for normal distribution, determining whether or not the assumption of normality is met, and multi-collinearity is evident. First, internal consistency for the variables measured at both Time 1 and Time 2 points were calculated. Second, the paired independent t-test was conducted to examine the longitudinal changes in college students’ PA motivation (perceived competence beliefs and task values), leisure-time PA participation, and depressive symptoms over one academic year, which stratified by sex groups.

Third, Pearson product-moment correlations were calculated among the variables measured at both time points stratified by sex groups. We then conducted two hierarchical regressions analyses to test the longitudinal predictive utilities of PA motivation and leisure-time PA participation on depressive symptoms (Time 2) after controlling all Time 1 measures and other covariates (i.e., sex, age, major, BMI—body mass index). Specifically, Time 1 variables (perceived competence beliefs-1, task values-1, leisure-time PA-1, and depressive symptoms-1) were entered at step 1; Time 2 variables (perceived competence beliefs-2 and task values-2, leisure-time PA-2) were entered at step 2 in the models; and the time 2 depressive symptoms served as the dependent variable in the models. Furthermore, a multivariate analysis of covariate (MANCOVA) was conducted to compare the mean differences on PA motivation and depressive symptoms between active (LSI ≥ 24) and inactive groups (LSI ≤ 23) based on Time 2 measures by controlling for Time 1 measures and other covariates (i.e., sex, age, major, BMI). Effect sizes including Cohen’s d, adjusted R-squared (aR^2^), and eta squared (η^2^) were reported. A statistically significant level of 0.05 in two-tailed tests was used across the data analysis.

## 3. Results

The examination of linear relationship among the variables suggested that the assumption of non-multicollinearity was met; the VIF and Tolerance statistics for the variables were all less than 5 and greater than 0.20 [33] (e.g., PA competence beliefs-1: 1.51 and 0.662; PA values-1: 1.505 and 0.664; leisure-time PA-1: 1.079 and 0.927; PA competence beliefs-2: 2.472 and 0.107; PA values-2: 1.899 and 0.527; leisure-time PA-2: 1.140 and 0.878; respectively). As shown in Table 2, results of analyses suggested that over a one-year period (from the beginning of freshman year to the beginning of sophomore year), both male and female students reported a significant increase in depressive symptoms (males: *M*_T1_ = 6.94 vs. *M*_T2_ = 8.47, Cohen’s *d* = 0.31; females: *M*_T1_ = 6.47 vs. *M*_T2_ = 7.75, Cohen’s *d* = 0.30). Both sexes displayed a low-to-moderate level of PA motivations in their first and second year of college, in which female college students reported a significant decrease in task values toward PA in their second year of college (*M*_T1_ = 3.47 vs. *M*_T2_ = 3.39, Cohen’s *d* = −0.13). Neither male nor female college students showed significant changes in leisure-time PA from first year to second year of college.

Results of correlation analyses (see Table 3) indicated that there were consistently positive associations between PA competence beliefs, PA values, and leisure-time PA for both Time 1 and Time 2 measures for both male and female groups (0.02 ≤ *rs* ≤ 0.27). PA competence beliefs and PA values in Time 1 and Time 2 had low but significant negative associations with depressive symptoms among both sex groups (−0.09 ≤ *rs* ≤ −0.24; *p* < 0.05).

Results also showed that while depressive symptoms were negatively associated with PA levels measured at both Time 1 and Time 2 among female college students, the negative relationship between PA and depressive symptoms was only observed for Time 2 measures among male college students.

The hierarchical regression model (Table 4) was conducted to examine the longitudinal associations of PA motivation and leisure-time PA with depressive symptoms (Time 2) for both male and female college students after controlling for PA motivation and leisure-time PA at Time 1 and covariates (i.e., age, major, and BMI). The results revealed that initial depressive symptoms and leisure-time PA (Time 1 measure; β = 0.45 and β = 0.08; *p* < 0.05, respectively) were significant predictors of depressive symptoms over time (aR^2^ = 23%) after controlling for all covariates among female college students. On the other hand, initial depressive symptoms (Time 1 measure) emerged as the consistent predictor of depressive symptoms at Time 2 over time in both female and male students (β = 0.29; β = 0.45; *p* < 0.01), respectively. It was found that PA competence beliefs at Time 2 significantly predicted depressive symptoms (Time 2 measure) among males (β = −0.17, *p* < 0.05; aR^2^ = 13%); but PA values emerged as the significant predictor of depressive symptoms (β = −0.11; *p* < 0.05; aR^2^ = 23%) among female college students regardless of their initial status of the depression level and demographic covariates.

The MANCOVA results (see Figure 1) revealed significant effects of PA groups (active group: *n* = 647, 64.4% vs. inactive group: *n* = 357, 35.6%) in PA motivations and depressive symptoms at Time 2 after controlling for Time 1 measures and other covariates (Wilks’ lambda = 0.977, *F*_(3, 976)_ = 7.79, *p* < 0.001, η^2^ = 0.023; observed power = 0.989). Active college students (leisure-time PA ≥ 24) reported significantly higher levels of PA competence beliefs (*F*_(1, 978)_ = 20.66, *p* < 0.001; η^2^ = 0.02) and PA values (*F*_(1, 978)_ = 9.71, *p* < 0.01; η^2^ = 0.01) but lower levels of depressive symptoms (*F*_(1, 978)_ = 3.93, *p* < 0.05; η^2^ = 0.004) when compared to their inactive peers (leisure-time PA < 24), regardless of gender.

## 4. Discussion

Guided by Lubans and colleagues’ conceptual model [18], the main purpose of the present study was to examine longitudinal associations between PA motivation, leisure-time PA participation, and depressive symptoms in a population of Chinese college students. The findings highlight the roles of psychosocial factors (i.e., PA competence beliefs in male, and PA values in female) play in influencing the level of depressive symptoms among college students regardless of their PA level (active or inactive). This study also observed high risks of experiencing depressive symptoms and low PA motivations among inactive college students compared to their active peers, suggesting the importance of promoting PA program and increasing PA opportunities during college years.

The results indicated that both male and female college students displayed significant increasing levels of depressive symptoms from their first year to second year of college. During the first year of college, students face many transitions and shifts of their life from adolescence to adulthood, including but not limited to different living and study environment, reduced family support, and increased financial burdens [15,16]. Early and easy access to regular mental health education and services for college students are crucial and strongly recommended during the entrance year of college. The constant low perceptions on competence beliefs and task values in physical activities among college students, especially females, point out the necessity to create a supportive climate to build up their positive PA perceptions (e.g., attitude, beliefs, values). Thus, healthcare personnel and PA instructors in higher education institutions may take note of the great value and need in making efforts to promote college students’ motivation toward PA and participation in regular PA in their leisure time.

Aligned with many previous studies, the correlational results indicated that college students’ leisure-time PA was significantly associated with their PA motivation across years [20,23,34]. The longitudinally tracked changes of PA motivation turned out to be the most salient predictors of depressive symptoms regardless of rate of PA participation and the initial status of depression level. Consistent with the hypothesized psychosocial mechanism in Lubans et al.’s conceptual model [18]; that is, when students believe they are competent and uphold perceived values toward PA, they are less likely to experience depressive symptoms over time. The fact that active college students demonstrated higher levels of PA motivation and leisure-time PA participation in the present study further illustrated that longitudinal association between PA and depressive symptoms could be consistently correlated with the level of PA motivation over time. This finding is also aligned with the findings in White et al.’s longitudinal study in that increased PA participation and PA motivation (i.e., competence beliefs, values) may help alleviate depressive symptoms among college students [23]. Meanwhile, these findings provide a valuable insight that improving college students’ competence beliefs and task values toward PA in both active and inactive college students are critical in order to prevent or reduce the prevalence of depressive symptoms.

One critical finding is that the psychosocial correlates towards depressive symptoms were different between female and male college students, in which perceived competence was more prominent in females while task values was more critical in males. This result extends the understanding of the Lubans et al. conceptual model as regards to sex differences [18]. This finding suggests that efforts to maintain or foster male college students’ competence beliefs and expectations for successful engagement in PA have the potential to alleviate or prevent depressive symptoms over time. Hence, school administrators and significant others (e.g., peers, parents, professionals) can create a supportive environment (e.g., offering group exercise club) to encourage PA participation for male college students. Previous research has concluded that support from significant others and supportive environment showed favorable effects on PA motivation and increased PA participation among college students, including those who experienced depressive symptoms [6,35]. The significance of task values in predicting depressive symptoms in females, on the other hand, suggested that health intervention emphasizing the values of PA (e.g., interest, important, usefulness) may have consistent influence on the reduction of depressive symptoms among female students. Administrators and teachers in higher education institutions should become the champion in advocating and reinforcing the value and benefits of PA participation, as well as building a supportive PA environment to promote active lifestyles, which could consequently reduce the risk of accelerated depressive symptoms during the college years.

It is important to note that, regardless of sex, active college students consistently demonstrated higher PA motivation and lower depressive symptoms and low leisure-time PA participation across the first year of college compare to inactive peers. It was also noticed that male students reported slightly higher leisure time PA, PA motivation and depressive symptoms over time but the difference did not reach the statistically significance in this sample. This finding is consistent with a recent longitudinal study [36] which analyzed 1892 undergraduate students from 15 universities in China. Followed for a four-year investigation, it was revealed that linearly increased depressive symptoms in the first two years in both male and females but there were no significant sex differences in depression and stress levels. Thus, maintaining an active lifestyle or promoting PA programs/opportunities (e.g., sports clubs, daily PE classes) are recommended strategies for both male and female college students in order to prevent the likelihood of increased depressive symptoms in young adulthood over time. Given that very few college freshmen with depressive symptoms sought help from mental health and counselling professionals, higher education administrators may consider promote and facilitate PA participation while indirectly, simultaneously mitigating depressive symptoms among college students, including offering courses teaching life-long PA knowledge, skills, and strategies; strengthening the PA-related facility accessibility and maintenance for students; and establishing an active campus by creating and encouraging PA-related activities, tournaments, and clubs.

Taken together, this study was one of the initial attempts to track changes in PA motivation and leisure-time PA participation over time and to examine the psychosocial correlates on depressive symptoms among a population of college students in China. The major strengths of this research include the large sample size, the study design strengthening internal validity (i.e., the longitudinal research design paired with the employment of controlling confounding variables), and establishing hypotheses based on a strong conceptual framework. Nevertheless, more research is needed to extend this research line from a broader social ecological perspective (e.g., built environment, social support).

Some limitations to be noted in the present study. First, the self-report measures may limit the objectivity of the data reported in the present study. Using subjective instruments to measure all variables, particularly the leisure-time PA participation and depressive symptoms status, may lead to respondent-related error and personal bias. Given the study’s large scope of sampling, however, objective measures of PA participation and clinical-based measures on depressive symptoms were not feasible due to time limits and cost concerns, which, nevertheless, are posted as limitations. Future research is warranted to consider having a subset sampling with multiple measures through objective electronic devices (accelerometers or pedometers) or direct interviews with the participants. Second, this study may limit generalizability of the research findings due to the concern that the variables were measured only at two time points. Thus, future research is needed to obtain multiple measures over three or four time points to more precisely identify temporal changes in PA motivation and leisure-time PA participation across time. Lastly, this study examined only a population of Chinese college students without identifying socio-economic status, thus, caution should be exercised when generalizing results of this study to college students of other nations given that variations in societal and cultural perceptions toward voluntarily identifying and reporting one’s mental health status may exist, as well as variations in instruments used for measuring the variables. Generalizability may also be limited to college students experiencing mental health disorders other than depressive symptoms.

## 5. Conclusions

Aligned with the conceptual framework proposed by Lubans et al. [18], this study revealed that PA motivation (i.e., PA competence beliefs in males and PA values in females) were associated with depressive symptoms among the Chinese college students. Additionally, this group of Chinese college students demonstrated an increasing level of depressive symptoms over the first 12-month of their college life. Active college students were more likely to maintain PA competence beliefs and PA values and have lower depressive symptoms over an academic year when compared to their inactive peers. The finding suggested that sex-specified intervention strategies should be targeted and implemented aimed to promote a healthy and active lifestyle among college students over time.

## Figures and Tables

**Figure 1 ijerph-18-05121-f001:**
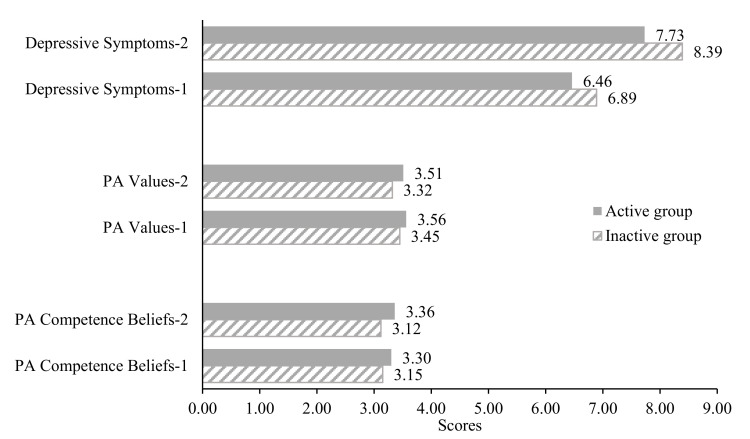
Active and Inactive Group differences in PA motivations and depressive symptoms over time. Note. Group Effect (Active vs. inactive): Wilks’ lambda = 0.977, *F*_(3, 976)_ = 7.79, *p* < 0.001, η^2^ = 0.023; observed power = 0.989). Time 1 measures and all covariates (i.e., age, gender, major, and BMI) were controlled.

**Table 1 ijerph-18-05121-t001:** Demographic characteristics of the sample (*n* = 1004).

Variables	*n* (%)	*M* (SD)	Min–Max
Age	1004		
School			
University-1	556 (55%)		
University-2	448 (45%)		
Sex			
Male	289 (28.5%)		
Female	725 (71.5%)		
Major			
Humanities	383 (38.1%)		
Social Science	394 (39.2%)		
Applied Science	145 (14.4%)		
Arts	82 (8.2%)		
Body Mass Index-T1		20.44 (2.91)	12.54–34.58
Height (Meter)-T1		1.66 (0.08)	1.43–1.88
Weight (Kilogram)-T1		56.25 (9.98)	35.00–103.00
Body Mass Index-T2		20.54 (2.92)	11.67–39.06
Height (Meter)-T2		1.66 (0.08)	1.43–1.88
Weight (Kilogram)-T2		56.62 (10.20)	31.00–105.00

**Table 2 ijerph-18-05121-t002:** The Results of the Paired *t*-test (*n* = 1004).

Group	Variables	Time 1	Time 2	*t*	*p*	*d*
		*M (SD)*	*M (SD)*			
Male (*n* = 284)	Depressive Symptoms	6.94 (3.90)	8.47 (4.50)	5.22	<0.001	0.31
	PA Competence Beliefs	3.34 (0.60)	3.37 (0.65)	1.10	0.27	0.07
	PA Values	3.66 (0.57)	3.58 (0.68)	−1.86	0.06	−0.11
	Leisure-time PA	37.96 (26.22)	37.08 (21.68)	−0.48	0.64	−0.03
Female (*n* = 720)	Depressive Symptoms	6.47 (3.95)	7.75 (4.36)	7.93	<0.001	0.30
	PA Competence Beliefs	3.21 (0.51)	3.23 (0.51)	1.11	0.27	0.04
	PA Values	3.47 (0.55)	3.39 (0.55)	−3.36	0.001	−0.13
	Leisure-time PA	30.91 (19.15)	30.95 (18.87)	0.05	0.96	0.002

PA = physical activity.

**Table 3 ijerph-18-05121-t003:** Correlations among study variables across Time 1 and Time 2 stratified by sex.

Measure	1	2	3	4	5	6	7	8
1. Depressive Symptoms-1	-	−0.11 **	−0.19 **	−0.05	0.46 **	−0.09 *	−0.13 **	−0.08 *
2. PA Competence Beliefs-1	−0.24 **	-	0.52 **	0.22 **	−0.11 **	0.66 **	0.36 **	0.17 **
3. PA Values-1	−0.17 **	0.63 **	-	0.20 **	−0.10 **	0.34 **	0.46 **	0.12 **
4. Leisure-time PA-1	−0.01	0.27 **	0.18 **	-	0.04	0.19 **	0.16 **	0.31 **
5. Depressive Symptoms-2	0.31 **	−0.24 **	−0.19 **	−0.05	-	−0.15 **	−0.15 **	−0.07 *
6. PA Competence Beliefs-2	−0.13 *	0.60 **	0.38 **	0.14 *	−0.24 **	-	0.55 **	0.26 **
7. PA Values-2	−0.10	0.43 **	0.44 **	0.08	−0.22 **	0.70 **	-	0.20 **
8. Leisure-time PA-2	0.00	0.12 *	0.02	0.17 **	−0.12 *	0.23 **	0.16 **	-

Note. Below diagonal is correlations in males; above diagonal is correlations in females. * = *p* < 0.05.; ** = *p* < 0.01.

**Table 4 ijerph-18-05121-t004:** Results of the regression analyses stratified by sex.

Independent Variables	Male Model				Female Model			
	R^2^	β	*t*	*p*	R^2^	β	*t*	*p*
**Step-1**	0.12 **				0.22 **			
Age		0.02	0.29	0.775		0.00	0.07	0.942
Major		−0.08	−1.42	0.157		0.00	−0.09	0.926
BMI-1		−0.05	−0.96	0.339		0.00	−0.12	0.907
Depressive Symptoms-1		0.28 **	4.87	0.000		0.46 **	13.53	0.000
Leisure-time PA-1		0.01	0.10	0.920		0.08 *	2.24	0.025
PA Competence Beliefs-1		−0.13	−1.77	0.078		−0.09 *	−2.21	0.027
PA Values−1		−0.04	−0.54	0.588		0.01	0.33	0.743
**Step-2**	0.13 **				0.23 **			
Age		0.02	0.33	0.744		0.01	0.25	0.802
Major		−0.09	−1.54	0.125		0.00	0.08	0.939
BMI-1		−0.07	−1.28	0.201		0.00	0.01	0.993
Depressive Symptoms-1		0.29 **	5.01	0.000		0.45 **	13.42	0.000
Leisure-time PA-1		0.00	0.01	0.994		0.08 *	2.39	0.017
PA Competence Beliefs-1		−0.03	−0.33	0.742		−0.07	−1.75	0.081
PA Values-1		−0.04	−0.55	0.583		0.05	1.23	0.220
BMI-2		0.11	1.15	0.251		0.10	1.38	0.169
Leisure-time PA -2		−0.08	−1.37	0.173		−0.04	−1.08	0.282
PA Competence Beliefs-2		−0.17 *	−2.44	0.015		−0.08	−1.65	0.100
PA Values-2		−0.09	−1.03	0.302		−0.11* *	−2.79	0.005

Note. BMI = body mass index; PA = physical activity; * = *p* < 0.05; ** = *p* < 0.01.

## Data Availability

The data presented in this study are available on reasonable request from the corresponding author. The data are not publicly available due to privacy restrictions.

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
