# Peer review of "Longitudinal Effects of Motivation and Physical Activity on Depressive Symptoms among College Students"

_ijerph, 2021, doi:10.3390/ijerph18105121_

Round 1

Reviewer 1 Report

The purpose of the study presented was to examine the relationship between physical activity motivation, self-reported leisure-time PA, and depressive symptoms among college students over time. The manuscript’s strengths include the sample size, longitudinal design, and alignment with Lubans conceptual framework. Weaknesses include self-reported leisure-time (that is included in the discussion section) and the lack of other demographic factors considered. I would like to see other demographic and education factors considered other than just age (e.g., socioeconomic measures, course load, college major).

Author Response

Thank you for your comments. We have carefully addressed your comments and all the changes made were highlighted in yellow in the text. Detailed responses have been attached here. 

Reviewer 2 Report

Thank you for the opportunity to review this study. Overall, I  thought it was well-designed and had clear outcomes. I have a few suggestions for enhancement:

  1. The title could include details of the participants, for example college students aged 18-22 years.
  2. Using the term 'college students' can mean different things in different countries. I would suggest that in addition the age range of participants is included.
  3. The abstract would benefit from 1-2 sentences at the beginning to set the scene/need to the study.
  4. The introduction draws on US physical activity recommendations but the study is in China. Is there more relevant literature that could be used?
  5. In the methods section there are no details of how the study was recruited to. Was it voluntary? In addition,  there are no details of how the measures were administered.
  6. The results section needs a table with the demographics of the participants.
  7. The discussion would benefit from being explicit in how the study addressed the study aims.  

Author Response

(The authors gave the same response as above.)

Reviewer 3 Report

This is an interesting paper and analysis. I have a few comments below. I feel that the relationships/findings described are over-simplified by the authors and need further clarification and more careful interpretation. It is not clear that the conclusions are supported by the findings and the conclusions need to more accurately reflect the findings of the study.

Comments:

Unclear why the abstract has the in-text citation – at least the (2016)

Please include p values in the abstract where appropriate

Please include a descriptives table at the start of results for the sample (male, female, all) and differences. Height, weight, BMI is important for the reader to understand the associations between weight, physical activity and depressive symptoms.

In the results, please make it clear what Mf and Ms means

In the data analysis section, it says “we conducted two hierarchical regressions” but in the table it is not clear that there were two steps. Is this the final model? What happened with the first step and did you enter depressive symptoms at time 1 in the first block?

Did you attempt to control for age or any other health conditions in your models?

Please also clarify what the time variable that you are controlling for is.

Can you please more clearly describe the rationale for a MANCOVA (as depressive symptoms seem to be your outcome of interest)? Please provide complete results with correct formatting (F statistic etc).

Different terms are used and it is confusing. In the methods you are looking at physical activity and motivation which contains two variables (perceived competence and task values based on the EVQ). Internal consistency of the motivation scale (which is physical activity motivation?) is good. But, then use two subscales. Is the reliability of the subscales also good? Why not just used physical activity and motivation based on the EVQ.

In the text results for Table 3, significant results are mentioned except the finding of leisure time 1 in females and depressive symptoms.

Line 272: Rather, inactive college students might display low PA motivation and this may be associated with greater symptoms of depression?

Line 301-305: What about reverse causality? Is this association between physical activity motivation and leisure time, rather than a relationship between physical activity motivation and depressive symptoms? Low motivation is itself (not just related to physical activity) is a feature of depression.

I think it needs to be more clearly (and reservedly stated) as to where depressive symptoms fit into the longitudinal relationship.

Line 318: The finding that male college students in your sample demonstrated higher depressive symptoms at both time points, in addition to higher (raw scores) competence, task value, and leisure time, has not been addressed.

Author Response

(The authors gave the same response as above.)

Reviewer 4 Report

Greetings Author(s),

Overall, this is a worthwhile manuscript.  This study investigates the longitudinal changes in PA motivation (perceived competence beliefs and task values toward PA), leisure-time PA participation, and depressive symptoms, and (2) the longitudinal associations between PA motivation, leisure-time PA participation, and depressive symptoms in a group of Chinese college students over a one year period.  The approach uses the Lubans et al. framework.  The research design and methods are appropriate.  Tables are supportive of the findings related in the manuscript.  The conclusions drawn from the study are supported by the data as detailed in the paper.

Additional considerations:

Line 46: Consider, It is well documented…

Line 50: Consider …meet daily PA recommendations…

Line 53: Recommend specifying the type of PA

Author Response

(The authors gave the same response as above.)

Reviewer 5 Report

Dear Authors, 

I really appreciated your work about the impact of motivation and physical activity on depressive symptoms. Given the persistent presence of depression or milder depressive symptoms among young people, this work raises important questions about the importance of mitigating these effects. For this, I consider the topic of your work extremely noteworthy, and I believe the relevance of this study.    
Despite this, there are some major concerns that should be addressed before the official publication in the International Journal of Environmental Research and Public Health.

Abstract: 

  • The authors immediately state the primary purpose of the study. Before this, I would suggest writing an additional sentence considering why it is important to investigate this topic.
  1. Introduction 
  • Even if Depression is a very known condition, I would suggest writing a sentence describing this clinical condition and its symptomatology. 
  • Line 54: what do the authors mean with “moderate”? Please try to define it.
  • Line 55 and 56: same clarification for “vigorous” and “moderate-to-vigorous
  • Line 61: The authors mentioned the phase between adolescence and adulthood and the consequences on people’s wellbeing and lifestyles. I think this aspect could be further discussed also with other references.  Since the authors also found differences in wellbeing during college, I think it is worth covering this phase more in detail. 
  • Line 101: When the authors talk about gender differences, I would suggest the inclusion also of this recent longitudinal work on gender differences: ”Gao, W., Ping, S., & Liu, X. (2020). Gender differences in depression, anxiety, and stress among college students: a longitudinal study from China. Journal of affective disorders, 263, 292-300.
  1. Materials and Methods

Participants: 

  • How was the sample size determined? Did you use a power analysis? Otherwise, which criteria did you follow to establish the size of your sample? Please specify this.
  • The authors described the sample providing information about age and academic field, however, no information considering ethnicity and socio-economic levels of the participants and/or their families was reported. The fact that the sample was formed by Chinese people is mentioned at the end, but I would suggest inserting this in the participant’s section. For what concerns socio-economic status, this information was not provided into the text and it should be added.

Procedures: 

  • More information about procedure should be given. Are the surveys completed online or in presence? The whole sample completed the surveys the same day or on different days? Which specific information are given to participants before the administration of the tests? What are their awareness’ levels considering the objectives of the study? Further information and details should be provided in this part.

Measures

  • I would suggest specifying in this part that all measures are self-administered.

Data analysis: 

  • Before the application of the paired independent t-test were the variables controlled for normality and homoscedasticity? Please specify this and mention the tests used for the checks. Otherwise, non-parametric tests should be applied. 
  • Further, when hierarchical regression analyses are performed it is also important to check for linearity of the variables and for the independence of residuals. Therefore, please report how the variables were controlled before the analysis.
  • In addition to this, in the implementation of multivariate analysis of covariate - MANCOVA, variables should also be checked for the possible presence of multicollinearity, in order to exclude the possibility for the dependent variables to be correlated to each other.
    Finally, post-hoc tests should be applied.
  1. Discussion
  • In general, given the limitations considering the sample of the current work, some general statements considering the findings should be milder and taken with caution. 
  • In the limitations part, another concern of the study that might prevent the generalization of the findings is the presence of a non-counterbalanced sample between males and females. For this, statements that take into account gender should be stated moderately.

Author Response

(The authors gave the same response as above.)

Round 2

Reviewer 3 Report

The authors have addressed all of the comments appropriately. I only have two minor suggestions: 

Line 25: Suggest space between M and age.

Lines 435-441: Thank you for adding this information. However, the sentence is very long and does not flow well. I suggest splitting into two sentences.

Reviewer 5 Report

Dear Authors, 

All my comments and suggestions have been adequately addressed and, therefore, I would suggest the Editor to take into account this important work and considering it for publication in International Journal of Environmental Research and Public Health.

Best,